# Ym1 protein crystals promote type 2 immunity

Ines Heyndrickx[1,2], Kim Deswarte[1,2], Kenneth Verstraete[3,4], Koen HG Verschueren[3,4], Ursula Smole[1,2], Helena Aegerter[1,2], Ann Dansercoer[3,4], Hamida Hammad[1,2], Savvas N Savvides[3,4], Bart N Lambrecht[1,2,5]*

[1]Laboratory of Immunoregulation and Mucosal Immunology, VIB-UGent Center for Inflammation Research, Ghent, Belgium; [2]Department of Internal Medicine and Pediatrics, Ghent University, Ghent, Belgium; [3]Unit for Structural Biology, VIB-UGent Center for Inflammation Research, Ghent, Belgium; [4]Department of Biochemistry and Microbiology, Ghent University, Ghent, Belgium; [5]Department of Pulmonary Medicine, Erasmus University Medical Center Rotterdam, Rotterdam, Netherlands

**Abstract** Spontaneous protein crystallization is a rare event, yet protein crystals are frequently found in eosinophil-rich inflammation. In humans, Charcot-Leyden crystals (CLCs) are made from galectin-10 (Gal10) protein, an abundant protein in eosinophils. Although mice do not encode Gal10 in their genome, they do form pseudo-CLCs, made from the chitinase-like proteins Ym1 and/or Ym2, encoded by *Chil3* and *Chil4* and made by myeloid and epithelial cells respectively. Here, we investigated the biological effects of pseudo-CLCs since their function is currently unknown. We produced recombinant Ym1 crystals which were shown to have identical crystal packing and structure by X-ray crystallography as in vivo native crystals derived from murine lung. When administered to the airways of mice, crystalline but not soluble Ym1 stimulated innate and adaptive immunity and acted as a type 2 immune adjuvant for eosinophilic inflammation via triggering of dendritic cells (DCs). Murine Ym1 protein crystals found at sites of eosinophilic inflammation reinforce type 2 immunity and could serve as a surrogate model for studying the biology of human CLCs.

*For correspondence:
Bart.Lambrecht@UGent.be

Competing interest: The authors declare that no competing interests exist.

## eLife assessment

This is an **important** and interesting account of the ability of Ym1 crystals to promote type 2 immunity in vivo, in mice. The data presented are **compelling**, building on and significantly advancing evidence this group has previously published on the type 2 immunogenicity of other protein crystals. The work will be of relevant interest to immunologists and researchers working on type 2 inflammatory disease, in the lung and in others tissues.

## Introduction

Protein crystals have been described already 150 years ago in eosinophilic inflammatory lesions in humans by *Charcot and Vulpian, 1860*; *Leyden, 1872*; *Aegerter et al., 2021*. Most proteins can only function properly in the soluble state and spontaneous protein crystallization in vivo remains a rare and largely unsolved scientific enigma (*Doye et al., 2004*; *Schönherr et al., 2018*).

Protein crystals have also been documented in mouse tissues, almost exclusively in the context of type 2 inflammation, rich in eosinophils, T helper 2 (Th2) cells and alternatively activated macrophages (*Guo et al., 2000*; *Hoenerhoff et al., 2006*; *Takamoto et al., 1997*; *Shultz et al., 1984*; *Harbord et al., 2002*; *Liu et al., 2009*; *Waern et al., 2010*; *Poczobutt et al., 2021*; *Fallon et al., 2001*; *Rehm et al., 1985*; *Green, 1942*; *Murray and Luz, 1990*; *Yang and Campell, 1964*; *Feldmesser*

et al., 2001; Baldán et al., 2008; Huffnagle et al., 1998; Mall et al., 2008). Such crystals in mice are colorless, but in histochemical stains they stain brightly with eosin, consistent with their protein content (Fallon et al., 2001). Before their biochemical nature was elucidated, they were therefore often referred to as eosinophilic crystals (Guo et al., 2000). These murine crystals are frequently observed in aged mice and genetically altered mice on a C57BL/6 or Sv/129 background, and in the context of eosinophil-rich asthma and parasitic infections (Guo et al., 2000; Roediger et al., 2015). Although eosinophilic crystals have been documented in the skin, biliary tree, stomach, lymph nodes, spleen and bone marrow of mice, it is clear that the principal site of crystal accumulation is in the lung. In several descriptions, accumulation of eosinophilic crystals was associated with severe lung inflammation and the disease was called eosinophilic crystalline pneumonia, acidophilic macrophage pneumonia, or crystalline pneumonitis (Hoenerhoff et al., 2006; Murray and Luz, 1990; Roediger et al., 2015; Henderson et al., 2002). In histology slides, the accumulated crystals are variable in size and shape ranging from needle-like crystals to rectangular plates (Fallon et al., 2001). Their resemblance to the needle-shaped human Charcot-Leyden crystals (CLCs) frequently resulted in the incorrect identification of these murine crystals as CLCs (Henderson et al., 2002; Zhu et al., 1999). However, only in 2000, almost 100 years after they were first described, Guo et al. purified crystals from the bronchoalveolar lavage (BAL) fluid of viable motheaten ($me^v/me^v$) mice and mass spectrometry analysis identified the chitinase-like protein (CLP) Ym1 as the sole protein component (Guo et al., 2000), thereby providing an unambiguous annotation of their content and distinguishing them from CLCs. Human CLCs are comprised of the eosinophil-derived galectin-10 (Gal10) protein, which is not encoded in the mouse genome (Leonidas et al., 1995; Swaminathan et al., 1999). Ym1 is mainly produced by alveolar macrophages, alternatively activated 'M2' macrophages, neutrophils and a subset of circulating pro-repair monocytes (Raes et al., 2002; Ikeda et al., 2018; Aegerter et al., 2022). In hyaline gastric lesions of aged 129S4/SvJae and B6,129 CYP1A2 null mice, pseudo-CLCs were found to consist of the closely related CLP Ym2, which is a protein mainly made by epithelial cells under the influence of IL-13 (Fallon et al., 2001; Zhu et al., 1999; Ward et al., 2001; Parkinson et al., 2021). Ym1 (encoded by the *Chil3* gene) and Ym2 (encoded by the *Chil4* gene) are CLPs belonging to the glycoside hydrolase family 18 (GH18) and are unique to mice, with no known human orthologues (Van Dyken and Locksley, 2018). Their potential receptors and natural binding partners are largely unknown, as are the exact functions of these still very enigmatic proteins.

We previously reported that recombinant Gal10 crystals produced in vitro showed crystal packing and structure identical to CLCs derived from patient mucus. When administered to the airways of mice, Gal10 crystals stimulated innate and adaptive immunity and acted as a type 2 adjuvant (Persson et al., 2019). Since mice do not encode the *LGALS10* gene, Gal10 has the potential to act as a neo-antigen in this setting. Here, we tested the in vivo relevance of pseudo-CLCs since these are made endogenously in mice at sites of eosinophilic inflammation, albeit mainly by macrophages and epithelial cells. We showed that recombinant in vitro made Ym1 and Ym2 crystals were structurally similar to crystals isolated from mice and to each other, and when administered to the airways of mice, crystalline Ym1, but not soluble Ym1 protein stimulated innate and adaptive immunity and acted as an adjuvant for Th2 driven eosinophilic airway inflammation and immunoglobulin secretion.

## Results

### Recombinant Ym1 and Ym2 crystals are structurally similar to native crystals ex vivo

Individual crystals were harvested from the BAL fluid of $me^v/me^v$ mice (carrying a defective protein tyrosine phosphatase *Ptpn6* gene) developing crystalline pneumonia (Guo et al., 2000) and C57BL/6J-Tg(CAG-Gal10)[Bla] mice (transgenically overexpressing Gal10 from the ubiquitous CAG promotor, manuscript in preparation), sensitized and challenged with the allergen house dust mite (HDM). Crystallographic studies using microfocus synchrotron X-rays showed that the crystals obtained from the BAL fluid of $me^v/me^v$ mice were Ym1 crystals, in accordance with a previous report (Guo et al., 2000). We had hypothesized that the crystals obtained from the BAL fluid of C57BL/6J-Tg(CAG-Gal10)[Bla] mice were CLCs containing Gal10. However, crystallographic characterization of these crystals revealed that they are Ym2 crystals. In parallel, we produced recombinant Ym1 and Ym2 protein in FreeStyle 293 F cells. Spontaneous crystallization was aided by low pH and low temperature (4 °C)

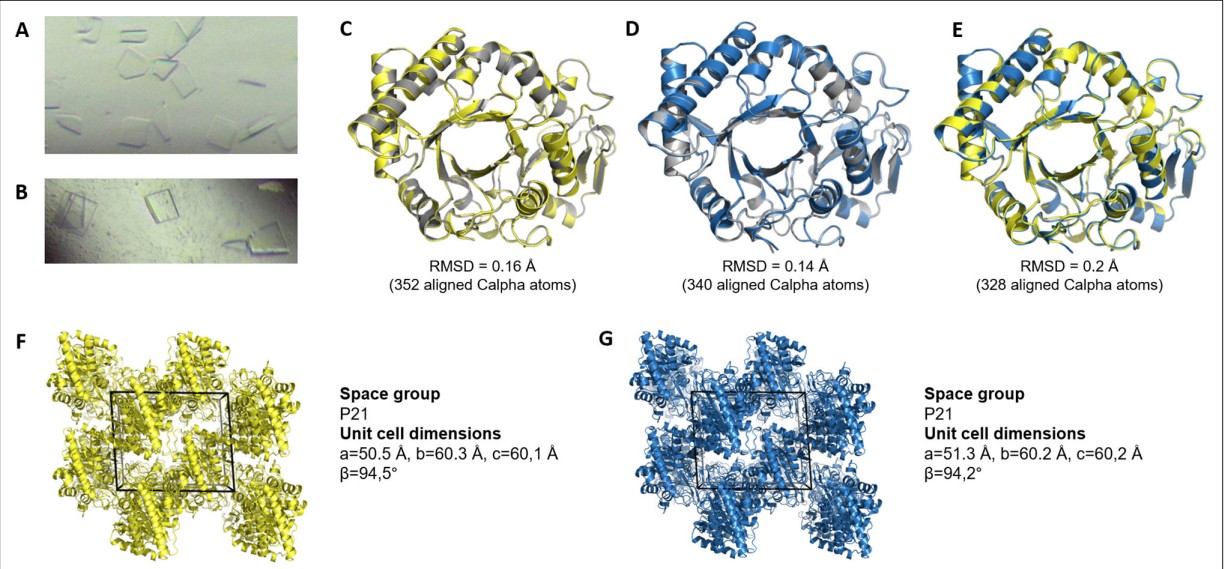

**Figure 1.** Recombinant Ym1 and Ym2 crystals are structurally similar to in vivo Ym1 and Ym2 crystals and to each other. (**A, B**) Light microscopy pictures of recombinant Ym1 (**A**) and Ym2 (**B**) crystals. (**C–E**) Structural superposition of the X-ray structures determined from a native in vivo grown Ym1 crystal isolated from the BAL fluid of me$^v$/me$^v$ mice (C+E) (yellow), a native in vivo grown Ym2 crystal isolated from the BAL fluid of a C57BL/6J-Tg(CAG-Gal10)$^{Bla}$ mouse (D+E) (blue) and a recombinant Ym1 (**C**) and Ym2 (**D**) crystal (grey). The low root-mean-square deviations (RMSDs) establish structural similarity. (**F, G**) Crystal packing of ex vivo Ym1 (**F**) and Ym2 (**G**) crystals revealing their crystallographic equivalence.

(*Figure 1A and B*). Native in vivo isolated Ym1 crystals and recombinant Ym1 crystals (*Figure 1C*) as well as native in vivo isolated Ym2 crystals and recombinant Ym2 crystals (*Figure 1D*) were crystallographically equivalent, sharing the same monoclinic space group (P21) and unit-cell parameters. Moreover, X-ray crystallographic studies showed that a native in vivo isolated Ym1 crystal and a native in vivo isolated Ym2 crystal are structurally similar to each other, which is not so surprisingly as Ym1 (*Chil3*) and Ym2 (*Chil4*) are two closely related proteins sharing 95% nucleotide sequence identity and 91% amino acid sequence identity (*Figure 1E, F and G*). Crystallographic data and refinement statistics can be found in *Supplementary file 1*. Thus, recombinant Ym1 and Ym2 protein crystals can be generated in vitro and are biosimilar to native Ym1 and Ym2 crystals.

## Ym1 crystals activate innate immunity

We next addressed whether Ym1 crystals could have pro-inflammatory effects in the lungs. Naïve WT C57BL/6 mice received an i.t. injection of 100 µg of endotoxin-low Ym1 crystals or control soluble Ym1 (*Figure 2A*). After 6 and 24 hr, there was a prominent influx of Ly6C$^+$ monocytes and eosinophils into the airways of mice receiving Ym1 crystals but not in those receiving soluble Ym1 or PBS. Neutrophil numbers were also increased in the Ym1 crystal group, but this failed to reach statistical significance (*Figure 2B*). This influx was accompanied by the production of the innate pro-inflammatory cytokines IL-6 and TNFα in BAL fluid (*Figure 2C*) and the inflammasome-dependent cytokine IL-1β and the alarmin IL-33 in lung tissue only in the group receiving crystalline Ym1 (*Figure 2D*). Other cytokines like IL-1α and granulocyte-macrophage colony-stimulating factor (GM-CSF) were not consistently induced in any of the groups (data not shown). The monocyte chemoattractant CCL2 and the eosinophil chemoattractant CCL24 (Eotaxin-2) were found enhanced in lung tissue of the crystal-treated group (*Figure 2E*), particularly at early time points, suggesting these paracrine signals were driving the cellular influx induced by crystals.

As we were unable to make a crystallization-deficient soluble Ym1 mutein, we used WT soluble Ym1 protein as a control. However, we took into consideration that soluble Ym1 might crystallize in vivo after i.t. injection. To test this, we stained BAL pellets using a polyclonal anti Ym1/2 antibody and analyzed the samples using the ImageStream flow cytometry platform. The results clearly showed that the i.t. injected pre-formed Ym1 crystals were still present as intact crystals after 6 and 24 hr. The soluble Ym1 protein indeed crystallized in vivo, with crystals being clearly detectable after 6 hr and

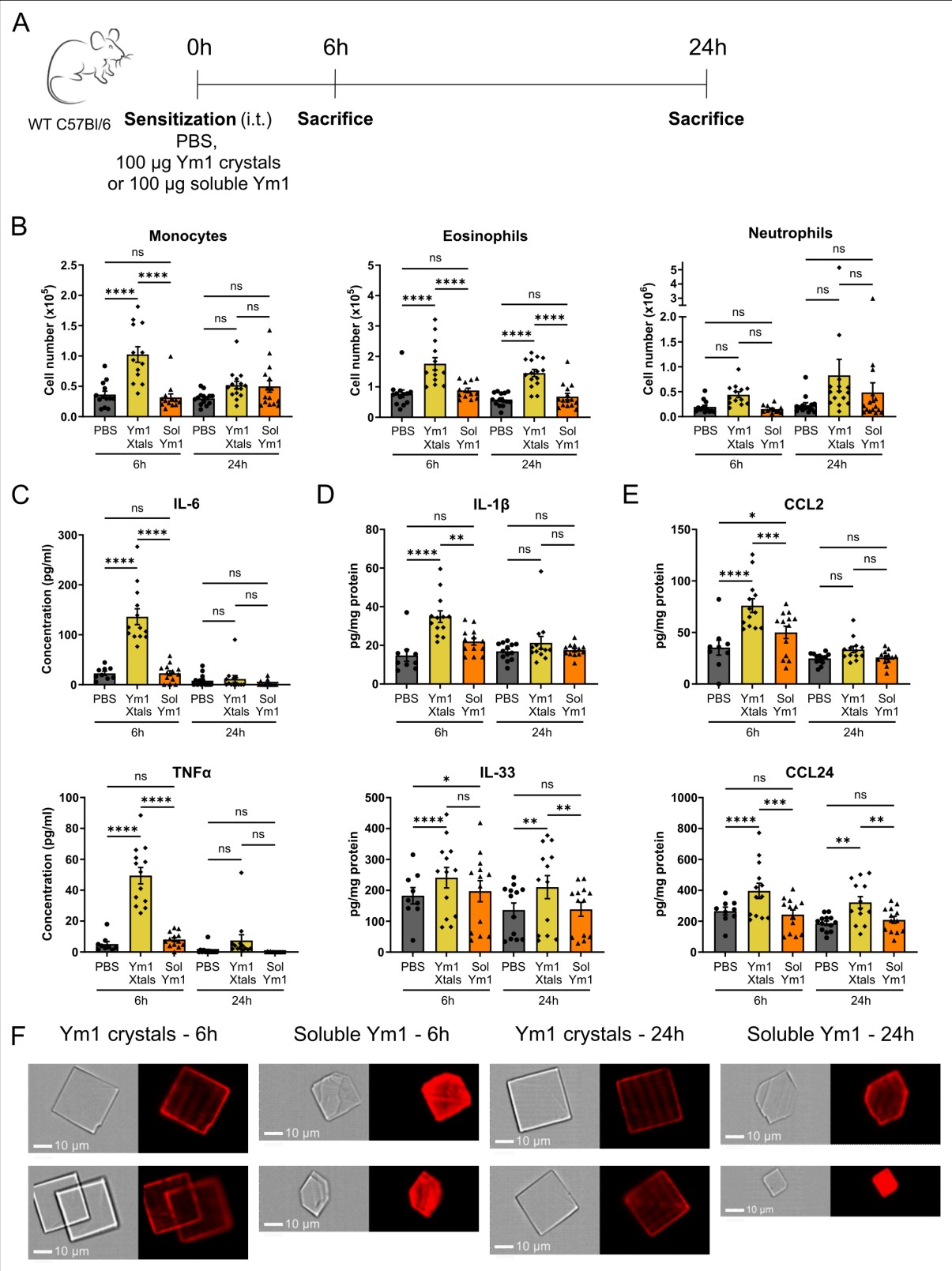

**Figure 2.** Ym1 crystals activate innate immunity. (**A**) Scheme representing the innate immune response experiment. (**B**) Flow cytometric analysis of the immune cell influx in the lungs of PBS, Ym1 crystal and soluble Ym1 treated mice measured 6 and 24 hr after i.t. injection. Data are pooled from 2 (for 6 hr time point) or 3 (for 24 hr time point) independent experiments with n=13, 13, 13, 14, 15 and 15, respectively. (**C**) Concentration of IL-6 and TNFα in the BAL fluid measured by ELISA. (**D, E**) Concentration of IL-1β, IL-33 (**D**), CCL2 and CCL24 (**E**) per milligram of protein of dispersed lung tissue

*Figure 2 continued on next page*

*Figure 2 continued*

measured by ELISA. Data are pooled from two independent experiments with n=13 per group. Data are shown as mean ± SEM. ns p≥0.05; *p<0.05; **p<0,01; ***p<0.001; ****p<0.0001. (**F**) ImageStream pictures of the BAL fluid of Ym1 crystal and soluble Ym1 treated mice 6 and 24 hr after i.t. injection. Samples were stained with an antibody recognizing Ym1/2 (red).

accumulating further after 24 hr. The crystals formed in vivo after i.t. injection of soluble Ym1 were also plate-like, but had a less uniform distribution and were smaller in size than the i.t. injected Ym1 crystals (*Figure 2F*). The total number of crystals found in the BAL pellet of soluble Ym1 treated mice (266 crystals in 5 pooled BAL pellets) was lower than in the BAL pellet of Ym1 crystal treated mice (1355 crystals in 5 pooled BAL pellets), potentially explaining why the soluble Ym1 group had less inflammation compared with the group receiving pre-formed Ym1 crystals.

## Ym1 crystals activate DCs in the lymph nodes

We next studied whether innate immunity triggered by Ym1 crystals was accompanied by the induction of adaptive immunity, particularly since IL-1β, IL-33, IL-6, and TNFα in the lungs can activate antigen presenting cells (*Lambrecht and Hammad, 2012*). Airway DCs bridge innate and adaptive immunity and can be activated by recombinant Gal10 crystals and monosodium urate (MSU) crystals (*Persson et al., 2019*; *Kool et al., 2011*). Different subsets of DCs can be clearly distinguished from recruited monocytes and resident macrophages based on the expression of CD26 and lack of expression of the complement C5a receptor CD88, and further separated into XCR1$^+$ cDC1 and CD172 (SIRPα)$^+$ cDC2 (see *Figure 3A* for gating strategy; *Aegerter et al., 2022*; *Rawat et al., 2023*). In mice given Ym1 crystals in the presence of fluorescently labeled OVA antigen (*Figure 3B*), there was a clear increase in the number of total lung-draining mLN CD26$^+$CD88$^-$CD11c$^+$CD64$^-$ conventional DCs (cDCs), numerically mainly caused by an increase in the number of CD172$^+$ cDC2, compared with mice receiving PBS or soluble Ym1, although XCR1$^+$ cDC1s also clearly reacted to the crystals. There was also an increase in the number of mLN CD88$^+$CD11b$^+$ monocytes that expressed intermediate levels of CD11c induced by crystals (*Figure 3C*). Not surprisingly, the exposure to crystalline Ym1 also led to an increase in cDC1s, cDC2s and monocytes taking up the co-administered fluorescent OVA antigen (*Figure 3D*). Exposure to soluble Ym1 and Ym1 crystals also induced an upregulation of the costimulatory molecules CD80 (data not shown) and CD86, and this effect was most pronounced with crystalline Ym1, and particularly in cDCs that were also taking up fluorescent OVA (*Figure 3E*).

## Ym1 crystals boost adaptive type 2 immunity

We next asked whether crystalline or soluble Ym1 could boost adaptive cellular immunity to co-administered OVA, when DCs that have taken up antigen and matured migrate to mLNs (*Figure 4A*). To evaluate induction of type 2 immunity in naïve antigen-specific T cells directly, cell tracer violet labeled OVA-reactive CD4$^+$ T cell receptor (TCR) MHCII-restricted transgenic T cells (OT-II T cells) crossed to the huCD2-IL-4 (KN2) reporter strain were transferred into WT mice, allowing us to track T cell division simultaneously with the acquisition of a Th2-polarized state in vivo (*Mohrs et al., 2001*). When innocuous OVA was injected i.t. together with Ym1 crystals, the adoptively transferred OT-II KN2 T cells showed increased proliferation 4 days (*Figure 4B*) and 7 days (data not shown) later in the mLNs, an effect absent when OVA was given with soluble Ym1. This led to the accumulation of higher numbers of OT-II T cells and Th2 polarizing KN2 +OT II T cells in the mLNs 4 days (*Figure 4C*) and 7 days (*Figure 4D*) after the i.t. administration. After 7 days, divided OT-II T cells and KN2 +OT II T cells could also be found in the lungs of mice receiving OVA mixed with Ym1 crystals (*Figure 4E*). These cells lost CD62L expression while expressing CD44, showing they had become effector cells (data not shown). These effects on T cell activation and Th2 polarization were again not observed when OVA was injected together with soluble Ym1. Early IL-4 production in OVA-reactive OT-II cells could also point to differentiation of T cells into T follicular helper (T$_{FH}$) cells that provide help for germinal center reactions. However, we did not add CXCR5 or PD1 to study T$_{FH}$ development in greater detail.

We finally tested whether Ym1 crystals could also promote allergic type 2 inflammation to inhaled harmless antigens, typically observed in asthmatics. WT C57BL/6 mice were sensitized i.t. with OVA mixed with Ym1 crystals on days 0 and 1 and received OVA i.n. challenges on days 11–13 (*Figure 5A*). These mice exhibited airway inflammation characterized by an influx of eosinophils, B cells and T cells (*Figure 5B*), a higher production of the type 2 cytokines IL-5, IL-10, and IL-13 in the supernatants of

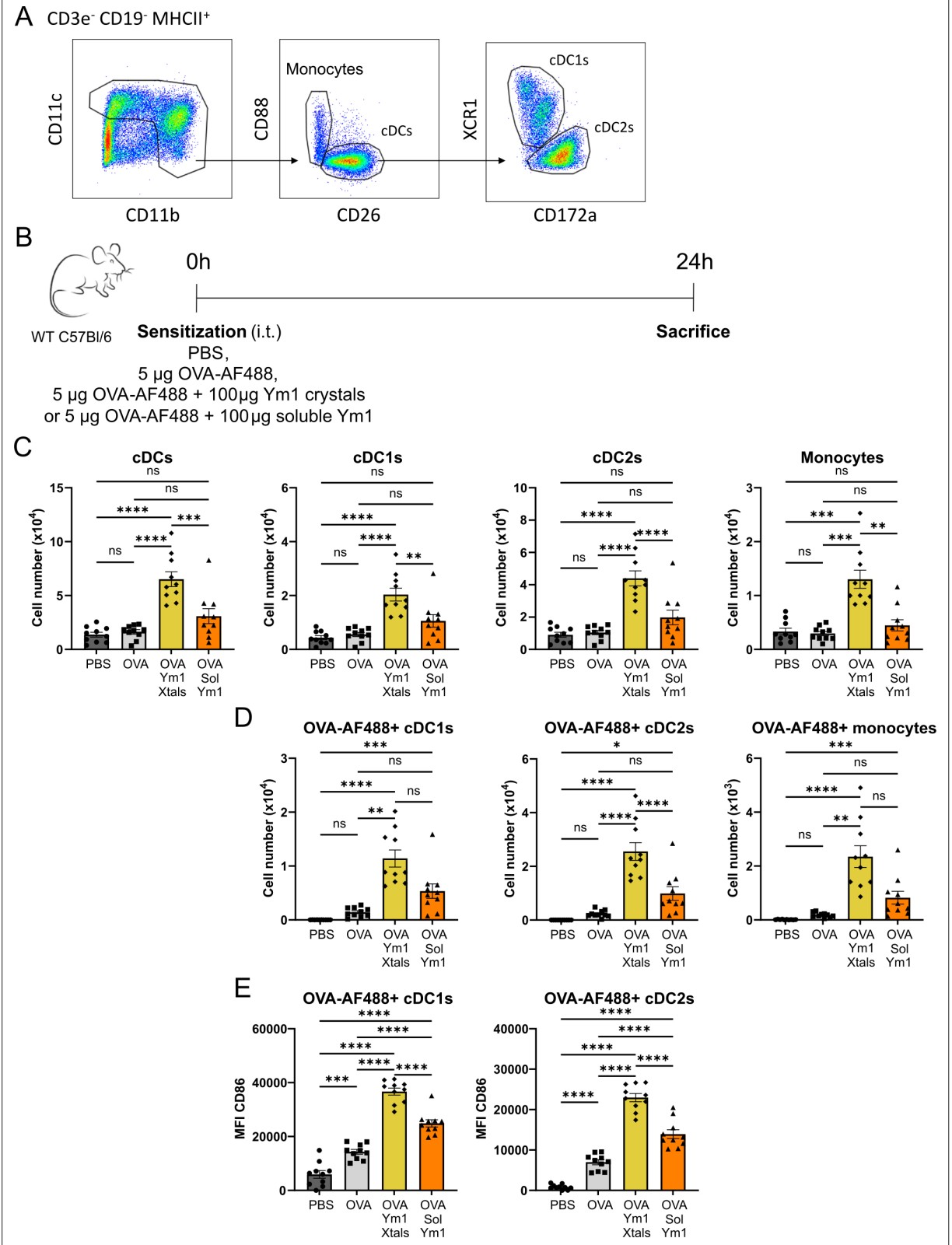

**Figure 3.** Ym1 crystals activate DCs in the lymph nodes. (**A**) Gating strategy to distinguish different subsets of DCs and recruited monocytes. (**B**) Scheme representing the DC activation experiment. (**C, D**) Flow cytometric analysis of the DC and monocyte influx in the mLNs of PBS, OVA-AF488 alone, OVA-AF488 plus Ym1 crystals and OVA-AF488 plus soluble Ym1 treated mice measured 24 hr after i.t. injection. (**E**) Median Fluorescence Intensity (MFI) of activation marker CD86 on OVA-AF488 +cDC1 s and cDC2s of PBS, OVA-AF488 alone, OVA-AF488 plus Ym1 crystals and OVA-AF488 plus soluble Ym1

*Figure 3 continued on next page*

*Figure 3 continued*

treated mice measured 24 hr after i.t. injection. Data are pooled from 2 independent experiments with n=10 mice per group. Data are shown as mean ± SEM. ns p≥0.05; *p<0.05; **p<0.01; ***p<0.001; ****p<0.0001.

mLN restimulation cultures (*Figure 5C*) and a robust OVA-specific IgG1 response, a hallmark of type 2 immunity, most likely driven by enhanced IL-4 production in Th2 or T$_{FH}$ cells (*Figure 5D*). These effects were not observed when OVA alone was used as the immunogen or when OVA was co-administered with soluble Ym1. Thus, the induction of allergic type 2 inflammation was specifically related to the crystalline state of Ym1. In contrast to recombinant Gal10 crystals, which mounted high titers of Gal10-specific IgG1 antibodies when used as adjuvant in the same experimental set-up (*Persson et al., 2019*), Ym1 crystals did not elicit an antibody response directed to Ym1 (*Figure 5E*). However, when mice received repetitive injections of Ym1 crystals in a different set-up via i.p. injection (*Figure 5F*), high titers of Ym1-specific IgM were detected (*Figure 5G*), suggesting that Ym1 crystals can trigger an autoimmune response or elicit a recall response from natural IgM antibody producing peritoneal B cells.

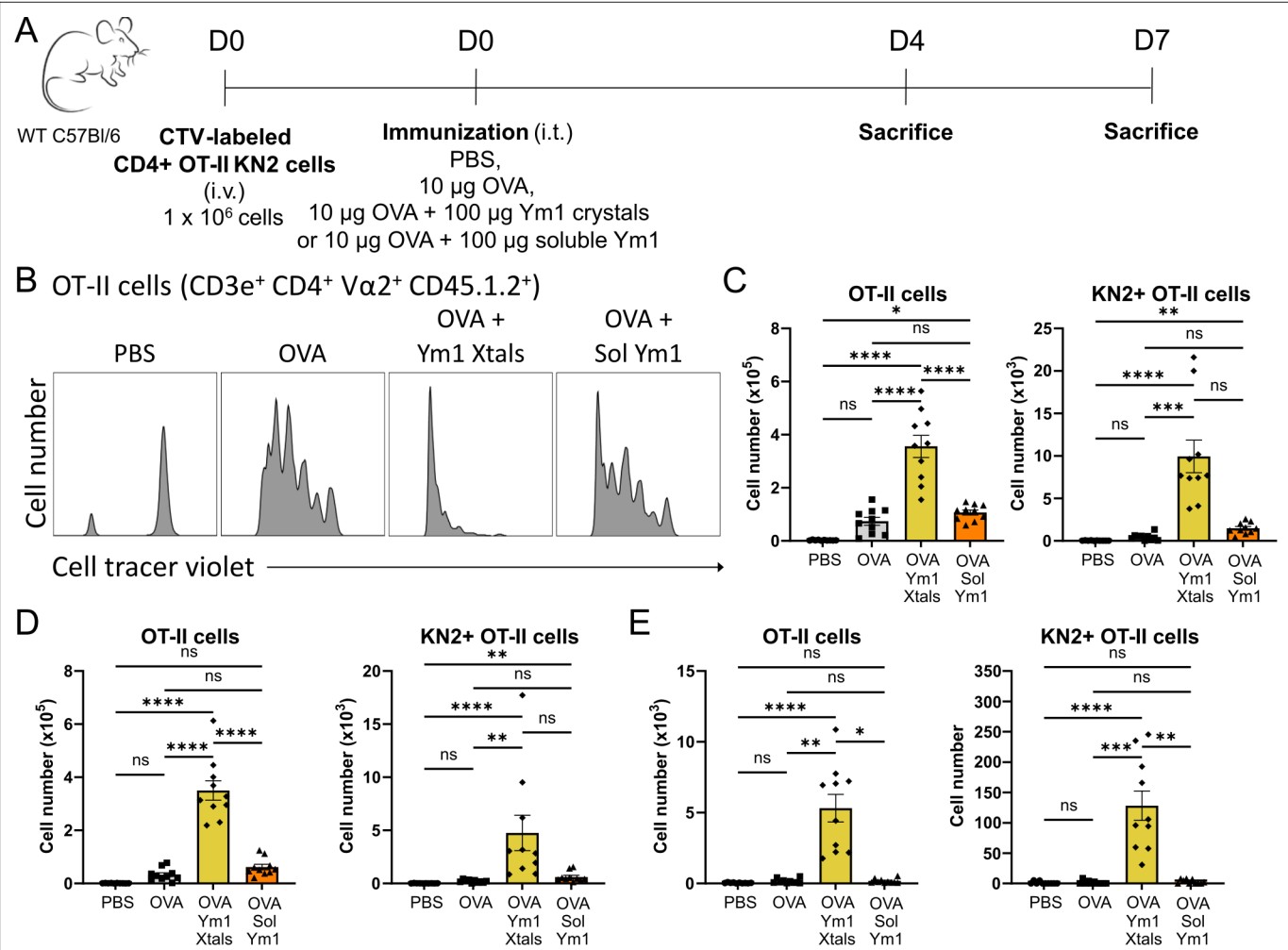

**Figure 4.** Ym1 crystals boost adaptive cellular immunity. (**A**) Scheme representing the used adoptive transfer model. (**B–E**) Proliferation and KN2 expression of OT-II T cells in the mLNs and lungs 4 and 7 days after i.t. administration of PBS, OVA alone, OVA plus Ym1 crystals or OVA plus soluble Ym1. (**B**) Representative histograms of the division profile of the adoptively transferred OT-II T cells (CD3e⁺ CD4⁺ Vα2⁺ CD45.1.2⁺) in the mLNs on day 4. (**C, D**) The numbers of OT-II T cells and KN2 +OT II T cells in the mLNs on day 4 (**C**) and day 7 (**D**). (**E**) The numbers of OT-II T cells and KN2 +OT II T cells in the lungs on day 7. Data are pooled from two independent experiments with n=10 mice per group. Data are shown as mean ± SEM. ns p≥0.05; *p<0.05; **p<0.01; ***p<0.001; ****p<0.0001.

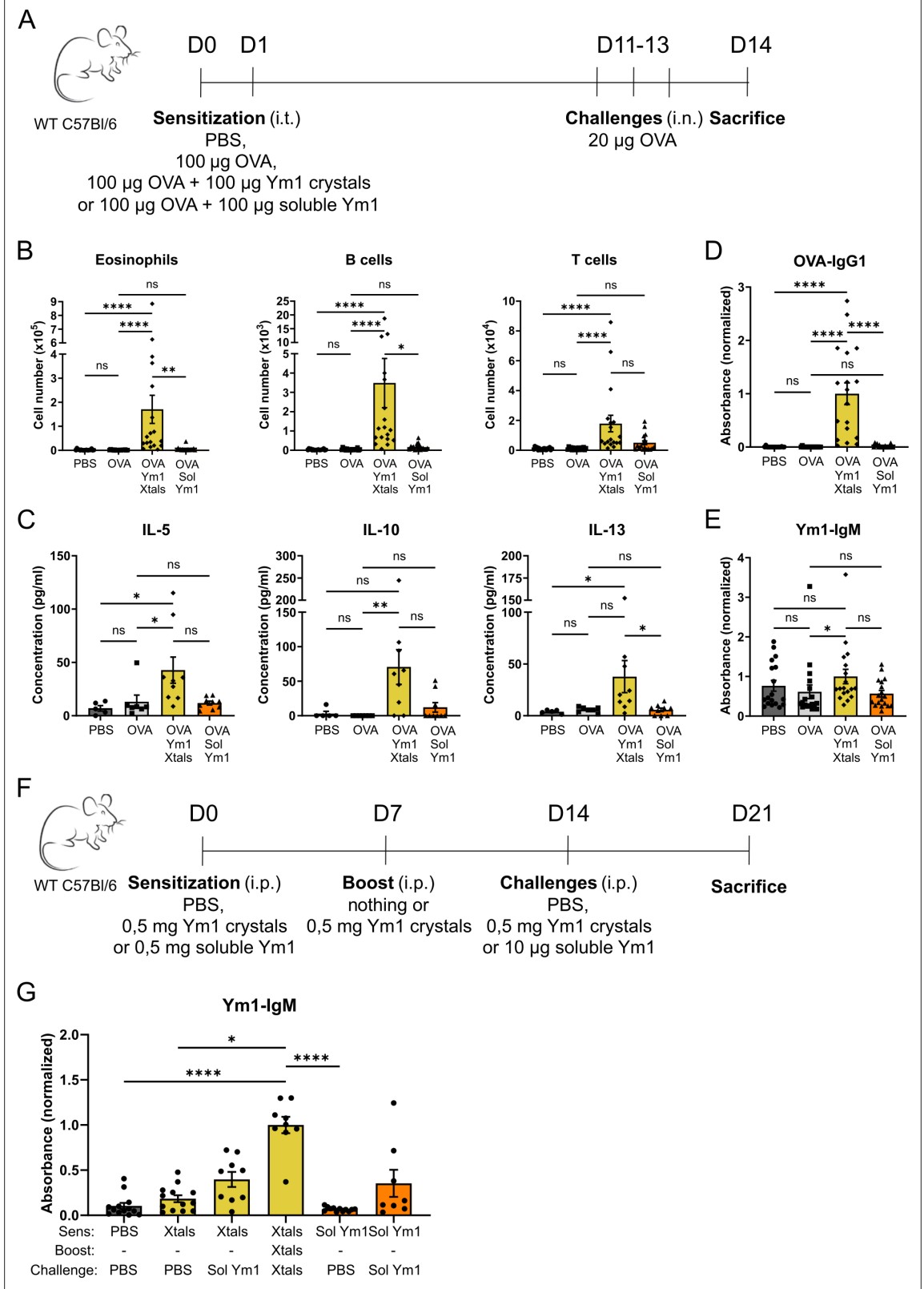

**Figure 5.** Ym1 crystals can act as type 2 adjuvant in OVA-induced asthma model and activate adaptive humoral immunity. (**A**) Scheme representing the used OVA-induced asthma model. (**B**) Flow cytometric analysis of the immune cell influx in the BAL fluid of mice sensitized i.t. with PBS, OVA alone, OVA plus Ym1 crystals or OVA plus soluble Ym1 and challenged i.n. with OVA. Data are pooled from two independent experiments with n=18 per group. (**C**) Levels of IL-5, IL-10, and IL-13 in the supernatant of mLN cells after restimulation with OVA for 20 hr. Data are from 1 experiment with n=5, 7, 9 and 9,

*Figure 5 continued on next page*

*Figure 5 continued*

respectively. (**D**) Serum OVA-specific IgG1 antibodies were measured by ELISA. (**E**) Serum Ym1-specific IgM antibodies were measured by ELISA. Data are pooled from two independent experiments with n=18 per group. (**F**) Scheme representing the adaptive humoral immune response experiment. (**G**) Serum Ym1-specific IgM antibodies were measured by ELISA. Data are pooled from three independent experiments with n=13, 13, 9, 9, 12, and 8, respectively. Data are shown as mean ± SEM. ns p≥0.05; *p<0.05; **p<0.01; ****p<0.0001. If pairwise comparison is not shown, the result is not significant.

## Discussion

Ym1 and Ym2 are among the most strongly induced proteins in the airways of mice after allergen exposure (*Jeong et al., 2005*; *Zhao et al., 2005*; *Zhao et al., 2007*; *Wong et al., 2007*; *Wong and Zhao, 2008*) and both proteins readily form pseudo-CLCs in vivo. As a result, Ym1 and Ym2 have become markers of type 2 inflammation. Here we show by ex vivo X-ray crystallography that Ym2 also crystallizes in vivo and that the phase transition of soluble Ym1/Ym2 to crystalline pseudo-CLC states leads to profound immunostimulatory effects on innate and adaptive type 2 immunity. Sutherland and colleagues have shown that vector-based overexpression of Ym1 or Ym2 in the lung increased the number of neutrophils in the BAL fluid (*Sutherland et al., 2014*). Whether these observed effects were due to the crystalline or soluble state of Ym1/2 remains unclear from these studies. Also in the present study, we lacked a proper soluble Ym1 control, since ImageStream analysis showed the soluble Ym1 protein to readily crystallize upon injection in the lungs. So far, several attempts to produce crystallization-deficient soluble Ym1 or Ym2 muteins failed, despite the fact that this was feasible for the CLC protein Gal10 (*Persson et al., 2019*). In *Persson et al., 2019*, we produced anti-Gal10 antibodies able to dissolve pre-existing CLCs and to inhibit key features of airway CLC crystallopathy in a humanized mouse model of asthma (*Persson et al., 2019*). Sutherland and colleagues have shown that blocking antibodies directed against Ym1/2 suppressed neutrophilic inflammation in the lungs of asthmatic mice and helminth-infected mice (*Persson et al., 2019*; *Sutherland et al., 2014*). However, it is unclear if these anti-Ym1/2 antibodies used suppressed lung neutrophilia by dissolving possible Ym1/2 pseudo-CLCs. We also observed that Ym1 crystals induced an auto-antibody response of the IgM isotype, suggesting intrinsic adjuvant activity in the crystalline state of Ym1. Antibodies to Ym1/2 have been described to occur also in a model of incipient neoplasia in mice overexpressing the Epstein Barr virus oncogene LMP1 in the skin, leading to chronic inflammation (*Qureshi et al., 2011*). However, it is unclear if induction of such auto-antibodies to Ym1/2 would be the result of spontaneous crystallization of Ym1/2 in the skin of these mice. Recognition of endogenously produced MSU or cholesterol crystals by IgM has been described previously (*Pilely et al., 2016*; *Kanevets et al., 2009*), and in the case of MSU, it was even proposed that IgM is required to turn soluble uric acid (UA) into an immunostimulatory adjuvant (*Kanevets et al., 2009*). Future studies will have to establish if IgM contributes in some way to the adjuvanticity of Ym1 crystals or facilitates the transition of soluble to crystalline Ym1.

Ym1 (*Chil3*) and Ym2 (*Chil4*) share 95% nucleotide sequence identity and 91% amino acid sequence identity and we now show here that native in vivo grown Ym1 crystals are structurally similar to native in vivo grown Ym2 crystals. Given these similarities, we assume that Ym2 crystals most likely will stimulate innate and adaptive immunity and act as a type 2 adjuvant, similar to Ym1 crystals, although we did not formally demonstrate this experimentally. We can only speculate why, from an evolutionary perspective, mice even express two very similar proteins that are expressed under similar conditions and share the peculiar property to crystalize in vivo. However, Ym1 is mainly expressed by macrophages, neutrophils and a subset of circulating pro-repair monocytes, while Ym2 is produced by lung epithelial cells, which could have discrete context dependent benefits. We have also not formally proven that Ym2 crystals have the same immunostimulatory effects on type 2 immunity as Ym1 crystals, but based on structural homology, we propose they do. Also, future studies will have to elucidate if soluble Ym1 and Ym2 have similar effects on immunity and tissue remodeling. A phase transition to its crystalline state may therefore mean different things for Ym1 and Ym2 biology in vivo.

Understanding the requirements for protein crystal formation during type 2 immunity, studying the mechanisms whereby protein crystals stimulate type 2 immunity and having the proper tools to be able to distinguish between the soluble and the crystalline state will help us to better understand if targeting CLCs in human asthma and chronic rhinosinusitis with nasal polyps (CRSwNPs) would be of value. Since CLCs and pseudo-CLCs are almost exclusively found in eosinophilic airway inflammation

across many species, and share many biological effects, it appears that protein crystallization is a more generalized feature of type 2 immunity.

The cellular influx and production of several cytokines and chemokines we observed after recombinant Ym1 crystal injection was also observed in earlier work where recombinant Gal10 crystals were injected into the lung of mice (*Persson et al., 2019*). Besides triggering innate immunity, Gal10 crystals, just like Ym1 crystals were able to recruit and activate DCs, activate cellular and humoral adaptive immune responses and act as type 2 adjuvants (*Persson et al., 2019*). Besides, not only protein crystals, but also organic non-proteinaceous crystals such as MSU crystals and crystalline alum were able to recruit and activate DCs and act as type 2 adjuvants (*Kool et al., 2011*; *Williams et al., 2014*), suggesting that promotion of type 2 response is a generic response to crystals of different chemical composition. The convergent evolution of CLCs and pseudo-CLCs as protein crystals in type 2 immunity strongly suggests a selective evolutionary advantage of these protein crystals in the enhancement of type 2-driven immune responses.

## Materials and methods

### Mice

Female wild type (WT) C57BL/6 mice were purchased from Janvier Labs (Saint-Berthevin, France), housed under specific pathogen-free conditions and used between 6 and 8 weeks of age. Investigators were not blinded to group allocation during experiments. No animals were excluded from the analysis. All experiments were approved by the animal ethics committee at Ghent University (EC2018-008, EC2019-088, EC2022-103, EC2023-057) and were in accordance with Belgian animal protection law.

### Production of recombinant Ym1 and Ym2 protein and crystals

A synthetic codon-optimized DNA sequence (Genscript) encoding murine Ym1 (Uniprot ID O35744, residues 22–398) was cloned in the pCAGG mammalian expression vector in frame with the mouse IgH signal peptide (MGWSCIIFFLVATATGVHS). A synthetic codon-optimized DNA sequence (Genscript) encoding murine Ym2 (Uniprot Q91Z98, residues 22–402) was cloned in the pTwist-CMV-BetaGlobin mammalian expression vector (Twist Biosciences) in frame with the mouse IgH signal peptide (MGWSCIIFFLVATATGVHS), and an N-terminal hexahistine-tag followed by a Tobacco Etch Virus (TEV)-protease cleavage site.

Ym1 and Ym2 were produced in FreeStyle 293 F cells in suspension. Cells were transfected with 1 µg/ml DNA using 2-fold excess of linear polyethylenimine (PEI; Polysciences). Conditioned medium containing the secreted recombinant protein was collected 4 days post transfection and filtered. Medium containing Ym1 was dialyzed to 20 mM Tris pH 8.0 and the Ym1 protein was purified via Q Sepharose anion exchange chromatography. Ym1 was further purified to a pure solution via size-exclusion chromatography using phosphate-buffered saline (PBS) as running buffer. Ym2 was purified from the clarified conditioned medium via immobilized metal affinity chromatography (IMAC) and size-exclusion chromatography with PBS buffer. Endotoxin levels were determined with an Endosafe-PTS system (Charles River) to be lower than 1 EU/mg of recombinant protein. To form recombinant Ym1 crystals, recombinant Ym1 protein (3–4 mg/ml) was incubated with 1 M Na-acetate buffer pH 4.6 with a buffer:target protein ratio of 1:10 (v/v). During a 24 hr incubation the protein solution was agitated a few times by inverting it five times, after which the solution turned cloudy due to the formation of protein crystals. The crystal-containing suspension was spun down (400 g for 5 min) and washed three times with sterile and endotoxin-free PBS before the crystals were resuspended in 1 ml of sterile and endotoxin-free PBS. The total protein concentration of this solution was determined by solubilizing 20 µl of the crystal suspension in an equal volume of 6 M guanidinium hydrochloride and measuring the absorbance at 280 nm with a Thermo Scientific NanoDrop 2000. The calculated extinction coefficient for Ym1 was 77155 $M^{-1}$ $cm^{-1}$.

### Crystallographic structure determination of ex vivo and recombinant Ym1 and Ym2 crystals

Single ex vivo Ym1 and Ym2 crystals were harvested from the BAL fluid of $me^v/me^v$ mice and C57BL/6J-Tg(CAG-Gal10)[Bla] mice by using mounted cryo-loops. Before being cryo-cooled in liquid nitrogen,

crystals were cryo-protected by a brief soak in PBS supplemented with 35% (v/v) ethylene glycol (Ym1) or 35% glycerol (Ym2). Recombinant Ym1 was crystallized in condition A7 of the Hampton Research Crystal Screen HT (0.1 M sodium cacodylate pH 6.5, 1.4 M sodium acetate) via vapor diffusion sitting drop crystallization experiments, and crystals were cryo-protected with mother liquor supplemented with 30% (v/v) glycerol. Following overnight incubation with TEV-protease, recombinant Ym2 was crystallized via vapor diffusion sitting drop crystallization experiments using 0.1 M sodium acetate pH 6.0 and 200 mM $CaCl_2$ as mother liquor, and crystals were cryoprotected with 30% (v/v) PEG-400. Diffraction experiments at 100 K were conducted on beamlines Proxima 1 (SOLEIL synchrotron, Saint-Aubin, France), P14 of Petra III (DESY, Hamburg, Germany) and X06SA (PXI) of the Swiss Light Source (Paul Scherrer Institute, Villigen, Switzerland). All data were integrated and scaled using the XDS suite (*Kabsch, 2010*) and the CCP4 suite (*Winn et al., 2011*). Molecular replacement was executed using Phaser (*McCoy et al., 2007*) employing the structure of Ym1 (pdb: 1vfb) as a search model. Further rebuilding of the model and refinement was performed in *COOT* (*Emsley et al., 2010*) and autoBuster (*Ferron et al., 2018*). Map and model validation was monitored throughout the refinement procedure using *COOT*, the CCP4 package and PDB_REDO server (*Joosten et al., 2014*).

## In vivo models

To analyze the innate immune responses, WT C57BL/6 mice were injected intratracheally (i.t.) with 100 µg Ym1 crystals or 100 µg soluble Ym1 diluted in 80 µl of PBS. After 6 and 24 hr, mice were euthanized by $CO_2$ inhalation or an overdose of pentobarbital (KELA Laboratoria) and BAL fluid and lungs were collected.

To analyze dendritic cell (DC) activation, WT C57BL/6 mice were injected i.t. with 5 µg Alexa Fluor 488-labeled ovalbumin (OVA-AF488) mixed with 100 µg Ym1 crystals or with 100 µg soluble Ym1 diluted in 80 µl of PBS. After 24 hr, mice were euthanized by $CO_2$ inhalation and mediastinal lymph nodes (mLNs) were collected.

To analyze the adaptive cellular immune responses, CD4 +OT II KN2 cells were purified from CD45.1.2 OT-II KN2 donor mice with the MojoSort Mouse CD4 T Cell Isolation Kit (Biolegend) and labeled with cell proliferation dye eFluor 450 (0.01 mM; Thermo Fisher Scientific) for 10 min in the dark at 37 °C. Next, $1\times10^6$ of these labeled cells (in 100 µl of PBS) were injected intravenously (i.v.) into WT C57BL/6 mice and subsequently, mice were injected i.t. with 10 µg ovalbumin (OVA, Worthington) or with 10 µg OVA mixed with 100 µg of Ym1 crystals or 100 µg soluble Ym1 (in 80 µl of PBS). For this experiment, OVA was pre-filtered through a 0.22 µm filter. On day 4 and 7, mice were euthanized by $CO_2$ inhalation and mLNs and lungs were collected.

To analyze the adaptive humoral immune responses, WT C57BL/6 mice were injected intraperitoneally (i.p.) on day 0 with 0.5 mg Ym1 crystals or 0.5 mg soluble Ym1, on day 7 with 0.5 mg Ym1 crystals and on day 14 with 0.5 mg Ym1 crystals or 10 µg Sol Ym1 all diluted in 500 µl PBS. On day 21 mice were euthanized with an overdose of pentobarbital (KELA Laboratoria) and blood was collected.

To analyze the type 2 adjuvanticity of Ym1 crystals, WT C57BL/6 mice were sensitized i.t. on day 0 and day 1 with 100 µg OVA (Worthington), 100 µg OVA mixed with 100 µg of Ym1 crystals or 100 µg OVA mixed with 100 µg soluble Ym1 diluted in 80 µl PBS. On days 11–13, all mice were challenged daily intranasally (i.n.) with 20 µg of OVA (Worthington) diluted in 40 µl of PBS. One day after the last challenge, on day 14, mice were euthanized with an overdose of pentobarbital (KELA Laboratoria) and blood, BAL fluid and mLNs were collected.

All i.t. and i.n. treatments were given after mice were anesthetized with isoflurane (2 liters/min, 2 to 3%; Abbott Laboratories).

Blood was obtained from the iliac vein and centrifuged at 10,000 rpm for 10 min at room temperature (RT). Serum was collected and stored at −20 °C until further use (serum antibody ELISA). BAL was performed by injecting PBS containing 0.01 mM EDTA (Lonza) through a tracheal canula. Subsequently, BAL fluid was centrifuged (400 g for 5 min at 4 °C). The supernatant was collected and stored at −20 °C until further use (cytokine ELISA) and the pellet was used for ImageStream or flow cytometry. To obtain lung and mLN single-cell suspensions for flow cytometry, lungs and mLNs were first cut with a scissor and then digested at 37 °C for 30 or 15 min respectively in RPMI-1640 (ThermoFisher Scientific) containing Liberase TM (20 µg/ml; Merck) and deoxyribonuclease (DNase) I (0.01 U/µl; Merck). The resultant cell suspensions were filtered through a 70 µm filter. To obtain mLN single-cell suspensions for flow cytometry or mLN restimulation cultures, mLNs were smashed on a 70 µm filter.

To measure pro-inflammatory cytokines, lungs were snap frozen in liquid nitrogen and homogenized with a TissueLyser II from Qiagen in tissue lysis buffer (40 mM tris-HCL [pH 6.8], 275 mM NaCl and 20% glycerol [Sigma-Aldrich]). One tablet of PhosSTOP (Sigma-Aldrich) and one tablet of Complete ULTRA tablets, Mini, Easypack (Sigma-Aldrich) were added per 10 ml of tissue lysis buffer. After homogenization, 2% IGEPAL CA-630 (U.S. Biological) was added. Subsequently, the samples were rotated for 30 min and centrifuged (20,000 × $g$ for 10 min at 4 °C). Supernatants were stored at −20 °C. The total protein concentration was measured with the NanoOrange protein quantitation kit from Thermo Fisher Scientific.

## Flow cytometry

Prior to staining for flow cytometry, single-cell suspensions were depleted of red blood cells (RBCs) by using RBC lysis buffer [0.15 M NH4Cl (Merck), 1 mM KHCO3 (Merck), and 0.1 mM Na2-EDTA (Merck) in MilliQ H2O] produced in-house. For all flow experiments, single-cell suspensions were incubated with the eBioscience Fixable Viability Dye eFluor 506 (ThermoFisher Scientific) to identify dead cells. Fc Block 2.4 .G2 (Bioceros) was used to block aspecific antibody binding. Cell surface markers were stained for 30 min at 4 °C in the dark. 123count eBeads Counting Beads (Thermo Fisher Scientific) were added to each sample to determine absolute cell numbers. Settings were calibrated using Ultra-Comp eBead Compensation Beads (Thermo Fisher Scientific). Data were acquired on an LSR Fortessa with FACS Diva software (BD Biosciences) and were analyzed with FlowJo software (Tree Star).

For the analysis of innate immune responses, lung single-cell suspensions were stained for flow cytometry using FITC-conjugated anti-Ly-6C (AL-21) (BD Biosciences), PE-conjugated anti-Siglec-F (E50-2440) (BD Biosciences), PE-Cy5-conjugated anti-CD3e (145–2 c11) (Thermo Fisher Scientific), PE-Cy5-conjugated anti-CD19 (1D3) (Thermo Fisher Scientific), PE-Cy7-conjugated anti-CD11c (N418) (Thermo Fisher Scientific), BD Horizon V450-conjugated anti-CD11b (M1/70) (BD Biosciences), Brilliant Violet 605-conjugated anti-CD45 (30-F11) (BD Biosciences), Alexa Fluor 647-conjugated anti-CD64 (X54-5/7.1) (BD Biosciences), Alexa Fluor 700–conjugated anti-Ly6G (1A8) (BD Biosciences) and APC-eFluor 780–conjugated anti-I-A/I-E (M5/114.15.2) (Thermo Fisher Scientific).

For the analysis of DC activation, mLN single-cell suspensions were stained for flow cytometry using PerCP-Cy5.5-conjugated anti-CD80 (16–10 A1) (BioLegend), eFluor 450-conjugated anti-CD11c (N418) (ThermoFisher Scientific), Brilliant Violet 650-conjugated anti-XCR1 (ZET) (BioLegend), Brilliant Violet 711-conjugated anti-CD64 (X54-5/7.1) (BioLegend), Alexa Fluor 647-conjugated anti-CD172a (P84) (BioLegend), Alexa Fluor 700- conjugated anti-Ly-6C (AL-21) (BD Biosciences), APC-eFluor 780-conjugated anti-MHCII (M5/114.15.2) (ThermoFisher Scientific), PE-conjugated anti-CD88 (20/70) (BioLegend), biotin-conjugated anti-CCR7 (4B12) (Thermo Fisher Scientific)+PE-CF594-conjugated streptavidin (BD Biosciences), PE-Cy5-conjugated anti-CD3e (145–2 c11) (Thermo Fisher Scientific), PE-Cy5-conjugated anti-CD19 (1D3) (Thermo Fisher Scientific), PE-Cy7-conjugated anti-CD86 (PO3) (BioLegend), BUV395-conjugated anti-CD11b (M1/70) (BD Biosciences) and BUV737-conjugated anti-CD26 (H194-112) (BD Biosciences).

For T cell response analysis, mLN and lung single-cell suspensions were stained using FITC-conjugated anti-Va2 TCR (B20.1) (BD Biosciences), Brilliant Violet 605-conjugated anti-CD45.1 (A20) (BioLegend), RedFluor710-conjugated anti-CD44 (IM7) (Thermo Fisher Scientific), APC-eFluor780-conjugated anti-CD25 (PC61.5) (Thermo Fisher Scientific), PE-conjugated anti-CD62L (MEL-14) (BD Biosciences), biotin-conjugated anti-hCD2 (RPA-2.10) (BioLegend) +PE-CF594-conjugated streptavidin (BD Biosciences), PE-Cy7-conjugated CD3e (145–2 C11) (BioLegend), BUV395-conjugated anti-CD4 (GK1.5) (BD Biosciences) and BUV737-conjugated anti-CD45.2 (104) (BD Biosciences).

BAL cells were stained with PE-conjugated anti-Siglec F (E50-2440) (BD Biosciences), PE-Cy5-conjugated anti-CD3e (145–2 c11) (Thermo Fisher Scientific), PE-Cy5-conjugated anti-CD19 (1D3) (Thermo Fisher Scientific), PE-Cy7-conjugated anti-CD11c (N418) (Thermo Fisher Scientific), BD Horizon V450-conjugated anti-CD11b (M1/70) (BD Biosciences), Alexa Fluor 700-conjugated anti-Ly6G (1A8) (BD Biosciences) and APC-eFluor 780-conjugated anti-MHCII (M5/114.15.2) (Thermo Fisher Scientific).

## ImageStream

For ImageStream analysis, the pellets obtained after centrifugation of the BAL fluid were pooled and stained for 30 min at 4 °C with unconjugated polyclonal goat anti-mouse Ym1/2 (AF2446, R&D

Systems; 1:100 diluted in PBS). After washing, the pellet was stained for 30 min at 4 °C in the dark with Alexa Fluor 647-conjugated polyclonal donkey anti-goat IgG (ThermoFisher Scientific; 1:1000 diluted in PBS). Fc Block 2.4 .G2 (Bioceros) was used to block aspecific antibody binding. Data were acquired on an Amnis ImageStreamX Mk II with INSPIRE software (Luminex) and were analyzed with IDEAS software (Luminex).

## Enzyme-linked immunosorbent assay (ELISA)

Interleukin (IL)–1β, IL-5, IL-6, IL-10, IL-13, IL-33, tumor necrosis factor alpha (TNFα) and C-C motif chemokine ligand 2 (CCL2) concentrations were evaluated using the Ready-SET-Go! ELISA kits (Thermo Fisher Scientific). CCL24 concentrations were evaluated using the Mouse CCL24/Eotaxin-2/ MPIF-2 DuoSet ELISA kit (R&D Systems). Serum Ym1-specific immunoglobulin M (IgM) and OVA-specific immunoglobulin G1 (IgG1) antibody levels were determined in flat-bottom 96-well plates (Greiner Bio-One), coated overnight with 0.1 mg/ml soluble Ym1 in PBS or with 0.1 mg/ml OVA (Worthington) in 0.1 M sodium carbonate buffer (pH 9.5), respectively. After washing and blocking for 1 hr, the samples diluted in PBS were added in duplicate and incubated for 2.5 hr at RT. After washing, biotinylated anti-mouse IgG1 or IgM detecting antibody (0.5 mg/ml; BD Biosciences) plus streptavi-din-HRP reagent (1:250; BD Biosciences) was added and incubated for 1 hr at RT. After another wash step, 1×TMB substrate solution (Thermo Fisher Scientific) was added and to stop the reaction, 2.5 N $H_2SO_4$ was used. Finally, the absorbance was read at 450 nm (and 650 nm as reference) with a Perkin Elmer multilabel counter and data were collected with Wallac 1420 Manager software (PerkinElmer).

## Statistical analysis

Experiments shown in *Figure 2* were analyzed with an ordinary two-way ANOVA test with Tukey's multiple comparisons test. For all experiments in *Figures 3–5*, the normality of each group was first checked by using the Shapiro-Wilk statistical test. Parametric data were analyzed with an ordinary one-way ANOVA test with Šidák's multiple comparison correction and nonparametric data were analyzed with a Kruskal-Wallis test with Dunn's multiple comparison correction. All statistical tests were performed in GraphPad Prism software (GraphPad Software).

## Acknowledgements

We acknowledge the crucial help of the staff of the VIB Protein Synthesis Core, the VIB Flow Cytometry core, the VIB Bioimaging core, and. SNS, KV, and KHGV thank the staff and management of synchrotron beamlines P14 (PETRA 3, Hamburg, Germany), Proxima 1 (SOLEIL, Saclay, France), and PX1 (Swiss Light Source, Villigen, Switzerland) for beamtime allocation and support during data collection of X-ray diffraction data. BNL acknowledges support from ERC advanced grant (789384 ASTHMA CRYSTAL CLEAR), a concerted research initiative grant from Ghent University (GOA, 01G010C9), a FWO Methusalem grant (01M01521) and a FWO EOS research grant (G0H1222N), and the Flanders Institute of Biotechnology (VIB). SNS acknowledges support to this project from the Ghent University GOA program (BOF17-GOA-028) and a FWO EOS research grant (G0H1222N).

## Additional information

### Funding

| Funder | Grant reference number | Author |
| --- | --- | --- |
| Fonds Wetenschappelijk Onderzoek | predoctoral FWO grant | Ines Heyndrickx |
| European Research Council | ASTHMA CRYSTAL CLEAR 789384 | Bart N Lambrecht |
| Ghent University | GOA grant 01G010C9 | Bart N Lambrecht |
| Fonds Wetenschappelijk Onderzoek | Methusalem grant 01M01521 | Bart N Lambrecht |

| Funder | Grant reference number | Author |
|---|---|---|
| Fonds Wetenschappelijk Onderzoek | EOS research grant G0H1222N | Savvas N Savvides Bart N Lambrecht |
| Ghent University | GOA grant BOF17-GOA-028 | Savvas N Savvides |

The funders had no role in study design, data collection and interpretation, or the decision to submit the work for publication.

### Author contributions

Ines Heyndrickx, Conceptualization, Formal analysis, Investigation, Visualization, Methodology, Writing - original draft, Writing – review and editing; Kim Deswarte, Ann Dansercoer, Investigation; Kenneth Verstraete, Conceptualization, Formal analysis, Investigation, Visualization, Methodology, Writing – review and editing; Koen HG Verschueren, Formal analysis, Investigation, Visualization; Ursula Smole, Helena Aegerter, Writing – review and editing; Hamida Hammad, Supervision; Savvas N Savvides, Conceptualization, Supervision, Methodology, Writing – review and editing; Bart N Lambrecht, Conceptualization, Supervision, Funding acquisition, Methodology, Writing – review and editing

### Author ORCIDs

Ines Heyndrickx http://orcid.org/0000-0001-8091-6390
Kim Deswarte http://orcid.org/0000-0002-9761-6973
Bart N Lambrecht http://orcid.org/0000-0003-4376-6834

### Ethics

All experiments conducted in this study were approved by the animal ethics committee at Ghent University (EC2018-008, EC2019-088, EC2022-103, EC2023-057) and were in accordance with Belgian animal protection law.

Reviewer #1 (Public Review): https://doi.org/10.7554/eLife.90676.3.sa1
Reviewer #2 (Public Review): https://doi.org/10.7554/eLife.90676.3.sa2
Author Response https://doi.org/10.7554/eLife.90676.3.sa3

## Additional files

### Supplementary files

• Supplementary file 1. Crystallographic data and refinement statistics. Key crystallographic data of ex vivo isolated Ym1 and Ym2 crystals for comparison with in vitro generated recombinant Ym1 and Ym2 crystals. Each dataset was collected from a single crystal. Values in parentheses are for highest-resolution shell.

• MDAR checklist

### Data availability

The recombinant Ym1 crystal structure, the ex vivo Ym1 crystal structure, the recombinant Ym2 crystal structure and the ex vivo Ym2 crystal structure have been deposited in the PDB under the accession codes 8P8Q, 8P8R, 8P8S and 8P8T, respectively.

The following datasets were generated:

| Author(s) | Year | Dataset title | Dataset URL | Database and Identifier |
|---|---|---|---|---|
| Verstraete K, Savvides SN, Lambrecht BN, Verschueren KHG, Heyndrickx I, Smole U, Aegerter H | 2023 | Recombinant Ym1 crystal structure | https://www.rcsb.org/structure/8P8Q | RCSB Protein Data Bank, 8P8Q |

*Continued on next page*

*Continued*

| Author(s) | Year | Dataset title | Dataset URL | Database and Identifier |
|---|---|---|---|---|
| Verstraete K, Savvides SN, Lambrecht BN, Verschueren KHG, Heyndrickx I, Smole U, Aegerter H | 2023 | Ex vivo Ym1 crystal structure | https://www.rcsb.org/structure/8P8R | RCSB Protein Data Bank, 8P8R |
| Verstraete K, Savvides SN, Lambrecht BN, Verschueren KHG, Heyndrickx I, Smole U, Aegerter H | 2023 | Recombinant Ym2 crystal structure | https://www.rcsb.org/structure/8P8S | RCSB Protein Data Bank, 8P8S |
| Verstraete K, Savvides SN, Lambrecht BN, Verschueren KHG, Heyndrickx I, Smole U, Aegerter H | 2023 | Ex vivo Ym2 crystal structure | https://www.rcsb.org/structure/8P8T | RCSB Protein Data Bank, 8P8T |

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
