## [Editor Report · eLife assessment]

This is an **important** and interesting account of the ability of Ym1 crystals to promote type 2 immunity in vivo, in mice. The data presented are **compelling**, building on and significantly advancing evidence this group has previously published on the type 2 immunogenicity of other protein crystals. The work will be of relevant interest to immunologists and researchers working on type 2 inflammatory disease, in the lung and in others tissues.

---

## [Referee Report · Reviewer #1 (Public Review)]

Summary:

The manuscript by Heyndrickx et al describes protein crystal formation and function that bears similarity to Charcot-Leyden crystals made of galectin 10, found in humans under similar conditions. Therefore, the authors set out to investigate CLP crystal formation and their immunological effects in the lung. The authors reveal the crystal structure of both Ym1 and Ym2 and show that Ym1 crystals trigger innate immunity, activated dendritic cells in the lymph node, enhancing antigen uptake and migration to the lung, ultimately leading to induction of type 2 immunity.

Strengths:

We know a lot about expression levels of CLPs in various settings in the mouse, but still know very little about the functions of these proteins, especially in light of their ability to form crystal structures. As such data presented in this paper is a major advance to the field.

Resolving the crystal structure of Ym2 and the comparison between native and recombinant CLP crystals is a strength of this manuscript that will be a very powerful tool for further evaluation and understanding of receptor, binding partner studies including ability to aid mutant protein generation.

The ability to recombinantly generate CLP crystals and study their function in vivo and ex vivo has provided a robust dataset whereby CLPs can activate innate immune responses, aid activation and trafficking of antigen presenting cells from the lymph node to the lung and further enhances type 2 immunity. By demonstrating these effects the authors directly address the aims for the study. A key apoint of this study is the generation of a model in which crystal formation/function an important feature of human eosinophilic diseases, can be studied utilising mouse models. Excitingly, using crystal structures combined with understanding the biochemistry of these proteins will provide a potential avenue whereby inhibitors could be used to dissolve or prevent crystal formation in vivo.

Generation of the crystal structure for Ym2 is a particular strength of the authors work and highlights the similarities between Ym1 and Ym2. Whilst the authors did not specifically examine Ym2 function, they have provided a discussion on this and speculate that Ym2 will function in a similar manner to Ym1.

The data presented flows logically and formulates a well constructed overall picture of exactly what CLP crystals could be doing in an inflammatory setting in vivo. Leaves open a clear and exciting future avenue (currently beyond the scope of this work) for determining whether targeting crystal formation in vivo could limit pathology.

Weaknesses:

It would have been nice for the authors to confirm whether Ym2 has similar functions to Ym1 using the in vivo and in vitro systems. However, they have discussed these points and raised it as a potential for future studies.

---

## [Referee Report · Reviewer #2 (Public Review)]

Summary:

This interesting study addresses the ability of Ym1 protein crystals to promote pulmonary type 2 inflammation in vivo, in mice.

Strengths:

The data are extremely high quality, clearly presented, significantly extending previous work from this group on the type 2 immunogenicity of protein crystals.

Weaknesses:

There are no major weaknesses in this study. It would be interesting to see if Ym2 crystals behave similarly to Ym1 crystals in vivo. Some additional text in the Introduction and Discussion would enrich those sections.

---

## [Author Response]

The following is the authors’ response to the original reviews.

**Public Reviews:**

**Reviewer #1 (Public Review):**
Summary:The manuscript by Heyndrickx et al describes protein crystal formation and function that bears similarity to Charcot-Leyden crystals made of galectin 10, found in humans under similar conditions. Therefore, the authors set out to investigate CLP crystal formation and their immunological effects in the lung. The authors reveal the crystal structure of both Ym1 and Ym2 and show that Ym1 crystals trigger innate immunity, activated dendritic cells in the lymph node, enhancing antigen uptake and migration to the lung, ultimately leading to induction of type 2 immunity.Strengths:We know a lot about expression levels of CLPs in various settings in the mouse but still know very little about the functions of these proteins, especially in light of their ability to form crystal structures. As such data presented in this paper is a major advance to the field.Resolving the crystal structure of Ym2 and the comparison between native and recombinant CLP crystals is a strength of this manuscript that will be a very powerful tool for further evaluation and understanding of receptor, binding partner studies including the ability to aid mutant protein generation.The ability to recombinantly generate CLP crystals and study their function in vivo and ex vivo has provided a robust dataset whereby CLPs can activate innate immune responses, aid activation and trafficking of antigen presenting cells from the lymph node to the lung and further enhances type 2 immunity. By demonstrating these effects the authors directly address the aims for the study. A key point of this study is the generation of a model in which crystal formation/function an important feature of human eosinophilic diseases, can be studied utilising mouse models. Excitingly, using crystal structures combined with understanding the biochemistry of these proteins will provide a potential avenue whereby inhibitors could be used to dissolve or prevent crystal formation in vivo.The data presented flows logically and formulates a well constructed overall picture of exactly what CLP crystals could be doing in an inflammatory setting in vivo. This leaves open a clear and exciting future avenue (currently beyond the scope of this work) for determining whether targeting crystal formation in vivo could limit pathology.Weaknesses:Although resolving the crystal structure of Ym2 in particular is a strength of the authors work, the weaknesses are that further work or even discussion of Ym2 versus Ym1 has not been directly demonstrated. The authors suggest Ym2 crystals will likely function the same as Ym1, but there is insufficient discussion (or data) beyond sequence similarity as to why this is the case. If Ym1 and Ym2 crystals function the same way, from an evolutionary point, why do mice express two very similar proteins that are expressed under similar conditions that can both crystalise and as the authors suggest act in a similar way. Some discussion around these points would add further value.

We agree with reviewer. We have further elaborated the discussion section including these points, stating clearly that more research needs to be done using Ym2 crystals before we can draw parallels in vivo.

Additionally, the crystal structure for Ym1 has been previously resolved (Tsai et al 2004, PMID 15522777) and it is unclear whether the data from the authors represents an advance in the 3D structure from what is previously known.

The crystal structure of Ym1 has indeed been previously solved, and we refer to that paper. In addition, we also provide the crystal structure of in vitro grown Ym1, ashowing biosimilarity. This, for the field of crystallography is a major finding, since it validates the concept that crystal structures generated in vitro can reflect in vivo grown structures. Moreover, the in vivo crystallization of Ym2 was unknown prior to this work, and is now clear as revealed by the ex vivo X-ray crystallography. The strength of our story is that we can now compare Ym1 and Ym2 crystals structures in detail.

Whilst also generating a model to understand Charcot-Leyden crystals (CLCs), the authors fail to discuss whether crystal shape may be an important feature of crystal function. CLCs are typically needle like, and previous publications have shown using histology and TEM that Ym1 crystals are also needle like. However, the crystals presented in this paper show only formation of plate like structures. It is unclear whether these differences represent different methodologies (ie histology is 2D slides), or differences in CLP crystals that are intracellular versus extracellular. These findings highlight a key question over whether crystal shape could be important for function and has not been addressed by the authors.

In contrast to the bipyramidal, needle-like CLC crystals formed by human galectin-10 protein (hexagonal space group P6522), the in vivo grown Ym1 and Ym2 crystals we were able to isolate for X-ray diffraction experiments had a plate-like morphology with identical crystallographic parameters as recombinant Ym1/Ym2 crystals (space group P21). We note that depending on the viewing orientation of the thin plate-like Ym1 crystals, they may appear needle-like in histology and TEM images. In addition, we can fully not exclude that both Ym1 or Ym2 may crystallize in vivo in different space groups (which could result in different crystal morphologies for Ym1/Ym2) but we have no data to support this. It is finally also a possibility that plate like structures can break up in vivo along a long axis as a result of mechanical forces, and end up as rod-or needle like shapes.

Ym1/Ym2 crystals are often observed in conditions where strong eosinophilic inflammation is present. However, soluble Ym1 delivery in naïve mice shows crystal formation in the absence of a strong immune response. There is no clear discussion as to the conditions in which crystal formation occurs in vivo and how results presented in the paper in terms of priming or exacerbating an immune response align with what is known about situations where Ym1 and Ym2 crystals have been observed.

Although Ym1 and Ym2 crystals are often observed in mice at sites of eosinophilic inflammation, they are not made by eosinophils, but mainly by macrophages and epithelial cells, respectively. In vitro, protein crystallization typically starts from supersaturated solutions that support crystal nucleation. Several factors such as temperature and pH can affect the solubility of Ym1 and Ym2 in vivo and thus affect the nucleation and crystallization process. For Ym1 and Ym2 we noticed in vitro that a small drop in pH facilitates the crystallization process. Although the physiological pH is 7.4, during inflammation, there is a drop in pH. This drop in pH is the result of the infiltration and activation of inflammatory cells in the tissue, which leads to an increased energy and oxygen demand, accelerated glucose consumption via glycolysis and thus increased lactic acid secretion. In addition, we cannot exclude that in vivo, the nucleation process for Ym1/Ym2 is facilitated by interaction with ligands in the extracellular space (e.g. polysaccharide ligands or other – yet to be identified – specific ligands to Ym1/Ym2).

**Reviewer #2 (Public Review):**
Summary:This interesting study addresses the ability of Ym1 protein crystals to promote pulmonary type 2 inflammation in vivo, in mice.Strengths:The data are extremely high quality, clearly presented, significantly extending previous work from this group on the type 2 immunogenicity of protein crystals.Weaknesses:There are no major weaknesses in this study. It would be interesting to see if Ym2 crystals behave similarly to Ym1 crystals in vivo. Some additional text in the Introduction and Discussion would enrich those sections.

We agree that this would be interesting to investigate, however, we choose to not include recombinant Ym2 crystal data in this report. However, we have further elaborated the discussion section including this point.

**Recommendations for the authors:**

**Reviewer #1 (Recommendations For The Authors):**
Suggestions for improved experiments and to strengthen findings:I think additional data on the ability of Ym2 crystals to induce an immune response would be advantageous. I'm not by any means suggesting the authors repeat all the experiments with Ym2 crystals, but even just the ability to show that Ym2 could promote type 2 immunity in the acute OVA model, would help to strengthen the argument that these crystals in general function in a similar way. Alternatively, a discussion on whether these protein crystals may function in different scenarios/tissues or conditions could help in light of additional data

We agree that this is an interesting point to investigate, however, we choose to not include recombinant Ym2 crystal data in this report. However, we have further elaborated the discussion section including this point.

Measuring IL-33 in lung tissue is difficult to interpret as cells will express intracellular IL-33 that is not active and may explain why the results in Fig 2D are not overly convincing. It could just be that Ym1 crystals are changing the number of cells expressing IL-33 (e.g macrophages, or type 2 pneumocytes) Did the authors also measure active IL-33 release in the BAL fluid which may give a better indication of Ym1's ability to activate DAMPs?

We also measured active IL-33 release in the BAL fluid, but due to the limited sample availability we could only measure this in one of the two repeat experiments, resulting in non-significant results for the BAL fluid. However, certainly for the 6h timepoint we saw a similar trend in the BAL fluid as in the lung tissue, meaning higher levels of IL-33 in the Ym1 crystal group compared to the PBS and soluble Ym1 group.

Crystals in Fig 2F staining with Ym1 appear to be brighter in the soluble Ym1 group. Is this related to increased packing of Ym1 in the crystals formed in vivo as opposed to those formed in vitro? Aside from reduced amount of crystals that form when you give soluble Ym1, could the type of crystal also be influencing the ability of soluble Ym1 crystals to generate an immune response?

Our X-ray diffraction experiments show that the packing of Ym1 is identical for in vivo and in vitro grown crystals. Possibly the apparent difference in brightness is caused by stochastic staining by the antibody. In this regard we note that the crystals formed from soluble Ym1 after 24h also can appear as less bright in a similar fashion as recombinant Ym1 crystals.

Overall, the data and writing of the manuscript is presented to a very high standardA few minor points:Fig 2F - a little unsure what the number in the left top corner of the images represented.

These numbers represent the picture numbers generated by the software, but as they don’t have any added value for the story, we removed these numbers from the images.

Not clear why two different expression vectors were used - one for Ym1 and one for Ym2?

Because we observed that recombinant Ym2 is more poorly secreted in the mammalian cell culture supernatant as compared to recombinant Ym1, we produced Ym2 with an N-terminal hexahistidine-tag followed by a Tobacco Etch Virus (TEV)-protease cleavage site to facilitate its purification.

**Reviewer #2 (Recommendations For The Authors):**
The authors briefly outline in their Introduction potential Sources of Ym1/2 in vivo, highlighting monocytes, M2 macrophages, alveolar macrophages, neutrophils and epithelial cells. Do DCs also make detectable/meaningful amounts of Ym1/2 in vivo, particularly in type 2 settings?

In the introduction we only highlighted the main cellular sources of Ym1 and Ym2, but there is literature available stating/showing that Ym1/2 is not only expressed by macrophages, neutrophils, monocytes and epithelial cells, but can also be induced in DCs and mast cells. We added the word ‘mainly’ to this sentence in the introduction, to make clear that macrophages, neutrophils and monocytes are not the only sources of Ym1.

Given the nicely demonstrated similarity of recombinant Ym1 and Ym2 crystals, I think it is important for the authors to include at least initial data on the outcome of recombinant Ym2 crystal admin to mice, in comparison to their Ym1 data.

We agree that this is an interesting point to investigate, however, we choose to not include recombinant Ym2 crystal data in this report. However, we have further elaborated the discussion section including this point.

Given the generation of crystals following in vivo administration of soluble Ym1, albeit at a lower level than when crystals were administered, it would be interesting to see if increased concentrations of soluble material show a dose dependent increase in lung inflammation readouts.

We agree that this would be an interesting point to investigate. Alongside this we could also titrate down the crystal dose, to see if there is a dose dependent decrease in lung inflammation readouts. However, at this time, we choose to not investigate this further.

I couldn't easily follow the authors' Discussion about potential ability of anti Ym-1/2 Abs to dissolve Ym1/2 crystals (similar to what they have demonstrated for Abs vs Gal10 crystals). Have they addressed this possibility experimentally? If so, addition of such data to the manuscript would be extremely interesting, given the obvious potential Ym1/2 crystal dissolving Abs for investigation of the role of these in a range of different murine models of type 2 inflammation.

We agree that the phrasing of this part of the discussion can be unclear/confusing. We rephrased this part to make it clearer. However, we did not address the possibility of Ym1/2 crystal dissolving antibodies experimentally.

In the Results section, the authors briefly comment on the pro-type 2 nature of Ym1 crystals in relation to their previous work with uric acid and Gal10 crystals, proposing that the pulmonary type 2 response may be a 'generic response to crystals of different chemical composition'. The Discussion would be enriched by deeper exploration of this comment.

We have further elaborated the discussion section including this point.